# Magnetic Otto Engine for an Electron in a Quantum Dot: Classical and Quantum Approach

**DOI:** 10.3390/e21050512

**Published:** 2019-05-20

**Authors:** Francisco J. Peña, Oscar Negrete, Gabriel Alvarado Barrios, David Zambrano, Alejandro González, Alvaro S. Nunez, Pedro A. Orellana, Patricio Vargas

**Affiliations:** 1Departamento de Física, Universidad Técnica Federico Santa María, Casilla 110-V, 2390123 Valparaíso, Chile; 2Departamento de Física, Universidad de Santiago de Chile (USACH), Avenida Ecuador 3493, 9170022 Santiago, Chile; 3Centro para el Desarrollo de la Nanociencia y la Nanotecnología, 8320000 Santiago, Chile; 4Departamento de Física, Facultad de Ciencias Físicas y Matemáticas, Universidad de Chile, Casilla 487-3, 8370456 Santiago, Chile

**Keywords:** magnetic cycle, quantum otto cycle, quantum thermodynamics

## Abstract

We studied the performance of classical and quantum magnetic Otto cycle with a working substance composed of a single quantum dot using the Fock–Darwin model with the inclusion of the Zeeman interaction. Modulating an external/perpendicular magnetic field, in the classical approach, we found an oscillating behavior in the total work extracted that was not present in the quantum formulation.We found that, in the classical approach, the engine yielded a greater performance in terms of total work extracted and efficiency than when compared with the quantum approach. This is because, in the classical case, the working substance can be in thermal equilibrium at each point of the cycle, which maximizes the energy extracted in the adiabatic strokes.

## 1. Introduction

The study of quantum heat engines (QHEs) [1] is focused on the search and design of efficient nanoscale devices operating with a quantum working substance. These devices are characterized by their working substance, the thermodynamic cycle of operation, and the dynamics that govern the cycle [2,3,4,5,6,7,8,9,10,11,12,13,14,15,16,17,18,19,20,21,22,23,24,25,26]. Among the cycles in which the engine may operate, the Carnot and Otto cycles have received increasing attention. In particular, the quantum Otto cycle has been considered for various working substances such as spin-1/2 systems [27,28] and harmonic oscillators [29], among others. Recently, an increasing number of experimental realizations for the quantum Otto cycle has been proposed in the literature [30,31,32,33]. Furthermore, it has been shown that thermal machines can be reduced to the limits of single atoms [34].

Previous studies of the quantum Otto cycle embedding working substances with magnetic properties have highlighted the role of degeneracy in the energy spectrum on the performance of the engine [35,36,37,38,39,40,41]. In this same framework, we highlight the work of Mehta and Johal [38], who studied a quantum Otto engine in the presence of level degeneracy, finding an enhancement of work and efficiency for two-level particles with a degeneracy in the excited state. In addition, Azimi et al. presented the study of a quantum Otto engine operating with a working substance of a single phase multiferroic $LiCu_{2}O_{2}$ tunable by external electromagnetic fields [39], which was extended by Chotorlishvili et al. [40] under the implementation of shortcuts to adiabaticity, finding an optimal output power for the proposed machine.

On the other hand, the classical description of the Otto cycle is characterized by state variables that are well-defined at each point of the cycle. In this sense, the main difference between the classical and quantum approach is that in the classical cycle the working substance can be at thermal equilibrium after each stroke. Classically, the adiabatic strokes are determined by the isentropic condition, which allows determining the state variables. For many systems, such as diamagnetic systems, which were considered in this study, the relation between the thermodynamics variables involved in the adiabatic stroke is not trivial in general and must be solved numerically [41].

In particular, it is interesting to compare the classical and quantum approaches for the same working substance and establish the conditions for each case appropriately. In this framework, several recent studies have focused on employing quantum coherence in the working fluid for enhancing the performance of the engine [42,43,44]. Recently, an interesting regime called “sudden cycles” [45] has been explored in an incoherent formulation avoiding off-diagonal elements of the density matrix, characterized by finite cooling power [46].

In this work, we study the classical and quantum performance of a multi-level Otto cycle in a diagonal formulation of the density matrix operator, where the working substance comprises a nanosized quantum dot under a controllable external magnetic field. This system is described by the Fock–Darwin model [47,48] that represents an accurate model for a semiconductor quantum dot. For this diamagnetic system, we find the point at which the quantum total work extracted becomes smaller than the classical one and we report, in the classical approach, an oscillating behaviour in the total work extracted that is not perceptible under the quantum formulation.

## 2. Model

Let us consider a system given by an electron in the presence of a parabolic potential and external magnetic field B. The Hamiltonian that describes the system is given by
(1)$$\begin{array}{c} \hat{H}=\frac{1}{2m^{*}}{\left(p+eA\right)}^{2}+U_{D}\left(x,y\right), \end{array}$$
where $m^{*}$ is the effective electron mass, A is the total vector potential, and the term $U_{D}\left(x,y\right)$ is given by
(2)$$\begin{array}{c} U_{D}\left(x,y\right)=\frac{1}{2}m^{*}\omega_{0}^{2}\left(x^{2}+y^{2}\right), \end{array}$$
which corresponds to an attractive potential describing the effect of the dot on the electron. The quantity $\omega_{0}$ is the parabolic trap frequency and can be controlled geometrically. If we consider a constant perpendicular magnetic field in the form
(3)$$\begin{array}{c} B=B\hat{z}\;, \end{array}$$
and the use of the vector potential A in the symmetric gauge (i.e., $A=\frac{B}{2}\left(-y,x,0\right)$), the solution of the eigenvalues of the Schrödinger equation are given by
(4)$$\begin{array}{c} E_{nm}=\hbar \Omega \left(2n+|m|+1\right)+\frac{1}{2}\hbar \omega_{c}m. \end{array}$$
where $\omega_{c}=\frac{eB}{m^{*}}$ is the cyclotron frequency, and *n* and *m* are the radial and magnetic quantum numbers (n=0,1,2,… and m=−∞,…,+∞), respectively. Ω is known as the effective frequency of the system corresponding to
(5)$$\begin{array}{c} \Omega =\omega_{0}{\left(1+{\left(\frac{\omega_{c}}{2\omega_{0}}\right)}^{2}\right)}^{\frac{1}{2}}. \end{array}$$

Notice that, when the parameter $\omega_{0}\to 0$, the energy levels of Equation (Equation 4) take the usual form of the Landau energy levels in cylindrical coordinates.

To obtain a more precise expression, especially when we consider the case of strong magnetic fields for the electron trapped in a quantum dot, we also take into account the electron spin of value $\frac{\hbar \hat{\sigma }}{2}$ and magnetic moment $\mu_{B}$, where $\hat{\sigma }$ is the Pauli spin operator and $\mu_{B}=\frac{e\hbar }{2m^{*}}$. Here, the spin can be in two possible states, either ↑ or ↓, with respect to the applied external magnetic field *B* in the z-axis. Therefore, we include the Zeeman term in the Fock–Darwin energy levels in Equation (Equation 4). Consequently, the energy spectrum is given by
(6)$$E_{n,m,\sigma }=\hbar \Omega \left(2n+|m|+1\right)+m\frac{\hbar \omega_{c}}{2}-\mu_{B}\sigma B.$$

The energy spectrum of Equation (Equation 6) is presented in Figure 1 for σ=−1 and σ=1. It is interesting to note that, for high magnetic fields $\left(\omega_{c}/2\omega_{0}>>1\right)$, things simplify in Equation (Equation 6) and we get the following expression:(7)$$E_{n,m,\sigma }=\frac{\hbar \omega_{c}}{2}\left(n+1/2+|m|+m\right)-\mu_{B}\sigma B\;,$$
where we observe that |m|+m=0 for m<0, therefore each Landau level labeled by *n* has infinite degeneracy.

In this paper, we consider a low-frequency coupling for the parabolic trap given by $\omega_{0}∼2.637$ THz which in terms of energy units corresponds to a coupling of approximately 1.7 meV. The selection of this particular value is to compare the intensity of the trap with the typical energy of intra-band optical transitions of the quantum dots [47]. The order of this transition is approximately around ∼1 meV for cylindrical GaAs quantum dots with effective mass given by $m^{*}∼0.067\;m_{e}$ [47,48,49].

For the classical approach, we employ the framework of Refs. [50,51,52,53], and, in particular, classical thermodynamic quantities for the Fock–Darwin model with spin can be obtained analytically using the treatment of Kumar et al. [54]. For a working substance in thermal equilibrium at inverse temperature $\beta =1/k_{B}T$, the partition function can be written as:(8)$$Z_{dS}=\frac{1}{2}\mathrm{csch}\left(\frac{\hbar \beta \omega_{+}}{2}\right)\mathrm{csch}\left(\frac{\hbar \beta \omega_{-}}{2}\right)\cosh\left(\frac{\hbar \beta \omega_{B}}{2}\right),$$
where the frequencies $\omega_{\pm }$ are:(9)$$\omega_{\pm }=\Omega \pm \frac{\omega_{c}}{2}.$$

Therefore, entropy (S(T,B)), internal energy (U(T,B)) and magnetization M(T,B) are simply given by
(10)$$S\left(T,B\right)=k_{B}\ln Z_{dS}+k_{B}T{\left(\frac{\partial \ln Z_{dS}}{\partial T}\right)}_{B},$$
(11)$$U\left(T,B\right)=k_{B}T^{2}{\left(\frac{\partial \ln Z_{dS}}{\partial T}\right)}_{B},$$
(12)$$M\left(T,B\right)=k_{B}T\left(\frac{\partial \ln Z_{dS}}{\partial B}\right).$$

Equations (Equation 10)–(Equation 12) are presented in Figure 2 for a parabolic trap corresponding to an energy of 1.7 meV together with the scheme of the Otto cycle that we consider. A very interesting behavior is observed for the entropy as a function of the magnetic field in Figure 2a. For external magnetic fields ≤1 T, the entropy decreases as the external field increases, but for values higher than 1 T we see the opposite behavior. This can be explained by the energy levels becoming closer to each other as the magnetic field increases, moving towards degeneracy. This behavior in the energy levels causes the entropy growth as the magnetic field increases. In addition, the change in the behavior of the entropy is affected by temperature, finding that the change of slope as a function of external magnetic field moves away from the 1 T value to higher values as we move to higher temperature of the working substance. This can be appreciated in Figure 2a. At the same time, the magnetization shows a crossing in its behavior as a function of magnetic field, as we can see in Figure 2b, where previous to this crossing at lower temperatures higher values of magnetization are obtained. This fact becomes essential for the total work extracted. In the cycle that we propose, the work is directly related to the change in the magnetization of the system as a function of magnetic field and temperature. On the other hand, we can see that the internal energy monotonically decreases in terms of the magnetic field for all temperatures considered. The reason for this is that the internal energy only depends on the derivative of $\ln Z_{dS}$ (see Equation (Equation 11)) with respect to temperature while the entropy has an additional term proportional to $\ln Z_{dS}$ (see Equation (Equation 10)) and the magnetization on its derivative with respect to the external field (see Equation (Equation 12)).

## 3. First Law of Thermodynamics and the Quantum and Classical Otto Cycle

The first law of thermodynamics in a quantum context has been discussed extensively in the literature. We follow the treatment in Refs. [50,51,52], which identifies the heat transferred and work performed during a thermodynamic process by means of the variation of the internal energy of the system.

First, consider a system described by a Hamiltonian, $\hat{H}$, with discrete energy levels, $E_{n,m,\sigma }$. The internal energy of the system is simply the expectation value of the Hamiltonian $E=\langle \hat{H}\rangle =\sum_{n}\sum_{m}\sum_{\sigma }p_{n,m,\sigma }E_{n,m,\sigma }$, where $p_{n,m,\sigma }$ are the corresponding occupation probabilities. The infinitesimal change of the internal energy can be written as
(13)$$\begin{array}{c} dE=\sum_{n}\sum_{m}\sum_{\sigma }\left(E_{n,m,\sigma }dP_{n,m,\sigma }+P_{n,m,\sigma }dE_{n,m,\sigma }\right), \end{array}$$
where we can identify the infinitesimal work and heat as
(14)$$\begin{array}{c} dQ:=\sum_{n}\sum_{m}\sum_{\sigma }E_{n,m,\sigma }dp_{n,m,\sigma },\;dW:=\sum_{n}\sum_{m}\sum_{\sigma }p_{n,m,\sigma }dE_{n,m,\sigma }. \end{array}$$

Equation (13) is a formulation of the first law of thermodynamics for quantum working substances. Therefore, work is then related to a change in the eigenenergies $E_{n,m,\sigma }$, which is in agreement with the fact that work can only be carried out through a change in generalized coordinates. It is important to note that the expressions of Equation (Equation 14) is only a particular case of the definition of work and heat for a case of a density matrix operator that is diagonal on the energy eigenbasis [52]. A more complete definition of Equation (Equation 14) can be found in Refs. [29,30,31,32,33,46].

The quantum Otto cycle is composed of four strokes: two quantum isochoric processes and two quantum adiabatic processes. This cycle can be seen in Figure 3, replacing the value of $S_{l}$ and $S_{h}$ for $P_{n,m,\sigma }\left(T_{l},B_{h}\right)$ and $P_{n,m,\sigma }\left(T_{h},B_{l}\right)$ in the vertical axis, respectively. For the cases that we consider, the quantum Otto cycle proceeds as follows.

1. Step B→A: Quantum adiabatic compression process. The systems, which is initialized in thermal equilibrium at temperature $T_{l}$, is isolated from the cold reservoir and the magnetic field is changed from $B_{h}$ to $B_{l}$, with $B_{h}>B_{l}$. During this stage the populations remain constant, so $P_{n,m,\sigma }\left(T_{l},B_{h}\right)=P_{n,m,\sigma }^{A}$. We remark that $P_{n,m,\sigma }^{A}$ does not yield a thermal state. No heat is exchanged during this process.

2. Step A→D: The system, at constant magnetic field $B_{l}$, is brought into contact with a hot thermal reservoir at temperature $T_{h}$ until it reaches thermal equilibrium. The corresponding thermal populations $P_{n,m,\sigma }\left(T_{h},B_{l}\right)$ are given by the Boltzmann distribution with temperature $T_{h}$. No work is done during this stage.

The heat absorbed for the working substance is given by
(15)$$\begin{array}{c} Q_{in}=\sum_{n}\sum_{m}\sum_{\sigma }\int_{A}^{D}E_{n,m,\sigma }dP_{n,m,\sigma }=\sum_{n}\sum_{m}\sum_{\sigma }E_{n,m,\sigma }^{l}\left[P_{n,m,\sigma }\left(T_{h},B_{l}\right)-P_{n,m,\sigma }^{A}\right], \end{array}$$
where $E_{n,m,\sigma }^{l}$ is the *n*-th eigenenergy of the system in the quantum isochoric heating process to an external magnetic field of value $B_{l}$.

3. Step D→C: Quantum adiabatic expansion process. The system is isolated from the hot reservoir, and the magnetic field is changed back from $B_{l}$ to $B_{h}$. During this stage the populations remains constant, thus $P_{n,m,\sigma }\left(T_{h},B_{l}\right)=P_{n,m,\sigma }^{C}$. Again, we remark that $P_{n,m,\sigma }^{C}$ is not a thermal state. No heat is exchanged during this process.

4. Step C→B: Quantum isochoric cooling process. The working substance at $B_{h}$ is brought into contact with a cold thermal reservoir at temperature $T_{l}$. Therefore, the heat released is given by
(16)$$\begin{array}{c} Q_{out}=\sum_{n}\sum_{m}\sum_{\sigma }\int_{C}^{B}E_{n,m,\sigma }dP_{n,m,\sigma }=\sum_{n}\sum_{m}\sum_{\sigma }E_{n,m,\sigma }^{h}\left[P_{n,m,\sigma }\left(T_{l},B_{h}\right)-P_{n,m,\sigma }^{C}\right], \end{array}$$
where $E_{n,m,\sigma }^{h}$ is the *n*-th eigenenergy of the system for an external magnetic field $B_{h}$.

The net work done in a single cycle can be obtained from $W=Q_{in}+Q_{out}$,
(17)$$\begin{array}{c} W=\sum_{n}\sum_{m}\sum_{\sigma }\left(E_{n,m,\sigma }^{l}-E_{n,m,\sigma }^{h}\right)\left(P_{n,m,\sigma }\left(T_{h},B_{l}\right)-P_{n,m,\sigma }\left(T_{l},B_{h}\right)\right), \end{array}$$
where we use the condition of constant populations along the quantum adiabatic strokes. Furthermore, the efficiency is given by
(18)$$\begin{array}{c} \eta =\frac{W}{Q_{in}}\;. \end{array}$$

The main difference between the classical and quantum Otto cycle is related to Points *A* and *C* in the cycle. In the classical case, the working substance can be at thermal equilibrium with a well-defined temperature at each point. On the other hand, for the quantum case, the working substance only reaches thermal equilibrium in the isochoric stages at Points *B* and *D*. After the adiabatic stages, the quantum system is in a diagonal state which is not a thermal state.

For the classical engine, the total work extracted by Equation (Equation 16) can be calculated by replacing $P_{n,m,\sigma }^{A}$ with $P(T_{A},B_{l})$ and $P_{n,m,\sigma }^{C}$ with $P(T_{C},B_{h})$, that is, it is obtained as a difference between the internal energy at adjacent points which can be calculated from the partition function
(19)$$Q_{in}=U\left(T_{h},B_{l}\right)-U\left(T_{A},B_{l}\right);\;Q_{out}=U\left(T_{l},B_{h}\right)-U\left(T_{C},B_{h}\right),$$
where $T_{A}$ and $T_{C}$ are determined by the condition imposed by the classical isentropic strokes. If we have the classical entropy, the intermediate temperatures $T_{A}$ and $T_{C}$ can be determined in two different forms:Finding the relation between the magnetic field and the temperature along an isentropic trajectory by solving the differential equation of first order given by
(20)$$dS\left(B,T\right)={\left(\frac{\partial S}{\partial B}\right)}_{T}dB+{\left(\frac{\partial S}{\partial T}\right)}_{B}dT=0,$$
which can be written as
(21)$$\frac{dB}{dT}=-\frac{C_{B}}{T{\left(\frac{\partial S}{\partial B}\right)}_{T}},$$
where $C_{B}$ is the specific heat at constant magnetic field.By matching two points within an isentropic trajectory
(22)$$\begin{array}{c} S\left(T_{l},B_{h}\right)=S\left(T_{A},B_{l}\right)\\ S\left(T_{h},B_{l}\right)=S\left(T_{C},B_{h}\right)\;, \end{array}$$
finding the magnetic field in terms of the temperature, throughout numerical calculation.

Therefore, from Equation (Equation 19) and $W=Q_{in}+Q_{out}$, the classical work is given by the difference of four internal energy in the form
(23)$$\begin{array}{c} W=U_{D}\left(T_{h},B_{l}\right)-U_{A}\left(T_{A},B_{l}\right)+U_{B}\left(T_{l},B_{h}\right)-U_{C}\left(T_{C},B_{h}\right), \end{array}$$

It is important to mention that the cycle operation in the counter-clockwise form starting at Point A described in Figure 2 gives negative work extracted, thus, to define a thermal machine correctly, we start the cycle at Point B, and we go through it in a clockwise direction. This is due to the particular behavior of the entropy as a function of magnetic field and temperature in the chosen zone marked with A, B, C and D. Therefore, the cycle described in the next subsection is the form of B→A→D→C→B and is presented in Figure 3.

The maximum values considered in our calculations for the temperatures and external magnetic field were 10 K and 5 T. Therefore, for the quantum cycle calculation (i.e., Equation (Equation 17)), we used the quantum numbers n=0 to n=10 and m=−33 to m=33 for Equation (Equation 6). The selection of this particular energy levels in this model is justified for the values of the thermal populations for the hot and cold temperatures of the reservoirs that we selected. Our numerical calculations indicated that the contributions of the other levels of energy can be neglected.

Finally, it is useful to express our results of total work extracted and efficiency in terms of the relation between the highest value ($B_{h}$) and the lowest value ($B_{l}$) of the external magnetic field over the sample. To do that, we used the definition of “magnetic length”, which is given by
(24)$$l_{B}=\sqrt{\frac{\hbar }{eB}},$$
allowing us to define the parameter
(25)$$r=\frac{l_{B_{l}}}{l_{B_{h}}}=\sqrt{\frac{B_{h}}{B_{l}}},$$
which represents the analogy of the compression ratio for the classical case. It is important to remember that the Landau radius is inversely proportional to the magnitude of the magnetic field. Therefore, for a major (minor) magnitude of the field, the Landau radius is smaller (bigger), and the *r* is well defined.

## 4. Results and Discussions

### 4.1. Classical Magnetic Otto Cycle

The condition given by Equation (Equation 21) (or Equation (Equation 22)) for the classical cycle give us information about the behavior of the external magnetic field and the temperature in the adiabatic stroke. In Figure 4a, we can appreciate the level curves of the entropy function S(T,B) and, Figure 4b shows some examples of isentropic strokes in a plot of S(B) vs. *B* for different temperatures. That example shows three cases of constant low (red-black curve, S=0.05), medium (yellow-black curve, S=0.10) and high (white-black curve, S=0.13) entropy. We observe in Figure 4a that there is a zone where the external field grows with the temperature of the sample and a zone where the opposite happens to maintain the entropy constant. At low working temperatures, the behavior changes near B=1 T, while as the temperature increases, the slope change occurs at higher values of the magnetic field, approaching B=2 T. Secondly, if we observe Figure 4b showing the case for S=0.13 (white-black line), we have a restricted area for field values lower to 3 T if we work with a maximum temperature of 10 K. Therefore, the movement of the magnetic field is not arbitrary if we think in a thermodynamic magnetic Otto cycle with two temperature reservoirs fixed at some specific values, more specifically, the reservoir associated to the hot temperature in the cycle. In addition, Figure 4 is the solution of S(T,B)= constant, obtained from the differential Equation (Equation 21) with different conditions (i.e., distinct values of the constant value of *S*). Therefore, Figure 4a depicts the entire family of solutions for the isentropic stroke of the engine of this particular system.

In our first example displayed in Figure 5, Point B has the value of the external field given by $B_{h}=4$ T and a temperature of $T_{B}=6.19$ K. The value of the temperature for Point D is fixed to $T_{D}=10$ K. Therefore, the Carnot efficiency of the proposal cycle is given by
(26)$$\eta_{Carnot}=1-\frac{T_{B}}{T_{D}}=1-\frac{6.19}{10}=0.381$$

Figure 5e shows different values of total work extracted (W) varying the value of $B_{D}$ from 4 T to 1.99 T. This variation in the external field is reflected in the movement of *r* in the form of $r=\sqrt{\frac{4}{B_{l}}}$. Therefore, *r* is in the range of 1≤r≤1.41. The parabolic trap is fixed to the value of 1.7 meV and the effective mass in the value of $m^{*}=0.067m_{e}$. In particular, Figure 5a–c shows the exact paths for the magnetic cycle for the maximum point obtained when multiplying W (Figure 5e) and the efficiency (η, Figure 5f). That point corresponds to r∼1.22 (black point in Figure 5d–f) and constitutes the best configuration of the systems to obtain the best W with the better η through the cycle. In addition, W and η are presented in Figure 5e,f for the optimal value of *r* parameter mentioned before. We observe that W obtained for that point is in the order of ∼0.038 meV with an efficiency of η∼0.28. We have corroborated the numeric result of total work extracted using the area enclosed by the cycle in Figure 5b of *M* versus *B*, as the work is W=−∫MdB [50,51,52] when the parameter changed during the operation of the engine in the external field. On the other hand, to obtain the solid lines presented in Figure 5d–f, we needed to make different cycles configuration keeping the values of the isothermal fixed as can be appreciated in the Appendix A (see the link after Section 5), made with the Mathematica software [55], where we show each shape that the cycle must have to generate a specific point of work. It is important to recall that we never reach the optimal value of η=0.381, i.e., the Carnot efficiency.

Due to the change of behavior in the entropy as a function of the external field, we obtained very interesting results for W when we explored the zone close to B=1 T. Before that point, the entropy decreases as function of the external field (B) and after that point entropy begins to increase. This fact can be used to explore the magnetic cycle in that zone finding an oscillatory behavior for W. In Figure 6, we show the cycle with operating temperatures $T_{B}=2.69$ K and $T_{D}=5.40$ K and external magnetic field moving between 2.995 T and 0.250 T and, consequently, the *r* parameter moving from 1 to 3.46. First, we observe a decreasing efficiency for r>1.75 in Figure 6f with a maximum value of η∼0.43 for r∼1.75. Therefore, also for this configuration, the Carnot efficiency ($\eta_{Carnot}∼0.5$ for this case) cannot be reached. Comparing these results with those previously discussed (when we avoid this particular region), we can see in Figure 5f that the efficiency asymptotically approaches to the efficiency of Carnot if we increase the intensity of the external magnetic field of the starting point of the cycle (Point B).

In Figure 6b, we can understand the oscillations in W interpreting these results using the expression W=−∫MdB. In Figure 6a–c, Points A–D correspond to the black point displayed in Figure 6d–f where we see that the work is still greater than zero but close to a vanishing situation. The reason there is still positive work at this point under study is that the total area enclosed to the right of the crossing point is larger than the other to the left. The magnetization presented in Figure 6b in the zone around the range of external magnetic field explored for this case (from 2.995 to 0.250T) clearly reverses his behavior and presents a crossing point close to B∼1.2 T for different temperatures. The area to the right of that point can be interpreted as a positive contribution to W while the left area contributes to negative work.

To explore if these oscillations in W are still obtained for higher temperature ranges, we plot in Figure 7 the work W for different values of $T_{D}$ with $T_{B}=2.69$ K fixed. We note that for higher temperatures than 7 K the oscillations found before disappear. It is only a reinforcement that the quantum effects of the working substance are only significant at low temperatures. On the other hand, as we expect, W grows as the difference between the temperature reservoir is larger, as shown in Figure 7a. However, for this case, the efficiency obtained is increasingly lower for increasingly larger temperature differences, as we can appreciate in Figure 7b.

### 4.2. Magnetic Quantum Otto Cycle

Next, we show the results of the evaluation of the quantum version of this magnetic Otto cycle for the same cases shown in Figure 5 and Figure 6. In Figure 8a, we plot the classical work (blue line) and the quantum work (red line) for the same sets of parameters in Figure 5. First, we note that the classical and quantum work are equal up to the value of r∼1.07. This means, for values close to the starting external magnetic field to Point B, we do not notice a difference between the classic and quantum formulation of the Otto cycle. As shown in Figure 8a, we found a transition from positive work to negative work not reflected in the classic scenario close to r∼1.26.

Additionally, we observe that the maximum positive value of the total work extracted for the quantum version of Otto cycle is reduced by approximately 0.01 meV compared to the classical counterpart. In particular, for the quantum version of this cycle, we did not found the oscillations in *W* presented in Figure 6e. Moreover, we found a transition from positive to negative work at some value of the *r* parameter. This is dramatically reflected in Figure 8b, where the absolute value of *W* is highly increased as compared with the classical approach.

In Figure 9, we present the work *W* per energy level and spin value for the most important values of our numerical calculations. We used the same parameter as in Figure 8b. We observed that the contribution given by σ=1 are positive up to *r* close to r∼1.6 being the energy levels $E_{0,-1}$, $E_{0,-2}$ and $E_{0,-3}$ those that contribute with the most positive values. Contrarily, for the case of σ=−1, we found that all contributions per energy level are negative. Therefore, the small region of positive work found in Figure 8b can only be associated to the spin up (σ=1) contributions.

To explore other operation regions for the magnetic Otto cycle, we calculated the total work extracted and efficiency for the same $\Delta T=T_{h}-T_{l}$ in a broad range of temperatures and the same $\Delta B_{max}=1.5$ T in different regions of the external magnetic field for the classical cycle and its quantum version. This is displayed in Figure 10 and Figure 11 where the dotted lines represent the classical results and the solid lines the quantum results. The three regions of temperature selected for these two figures are 1–4 K (blue lines), 4–7 K (black lines) and 7–10 K (red lines). First, we treat the case of $B_{B}=3.5$ T and $B_{D}$ moving from 3.5 T to 2.0 T in Figure 10, where we note large differences between the classical and quantum results for *W*, as can be seen in Figure 10b. On the other hand, if we observed the region of $3.5\leq B_{D}\leq 5.0$ T for a $B_{B}$ fixed, as shown in Figure 11b. The work and efficiency for the region of 1–4 K and 4–7 K present similar behavior for the classical and quantum versions. Only the case of 7–10 K shows a larger difference between this two approaches. For the case of the efficiency, we note in Figure 10c a major difference between the classical results and quantum results compare with the presented in Figure 11c and this is consistent with the reported results for the work *W*.

Summarizing, our results show that it is the classical engine case with larger total work extracted and efficiency compared to the quantum formulation. This can be explained as follows.

The main difference between the classical and quantum version of Otto cycle lies in the fact that, in the classical formulation, the working substance can be in thermal equilibrium at each point in the cycle. In the quantum approach, the working substance is a single system that can only be in a thermal state after thermalizing with the reservoirs, which happens only in the isochoric strokes. After the adiabatic strokes, the working substance is in a diagonal state which is not a thermal state. In our case, the non-thermal points for the quantum case are Points C and A in Figure 3. The quantum work given by Equation (Equation 17) can be rewritten in the convenient form
(27)$$W=U_{D}\left(T_{h},B_{l}\right)-\sum_{n,m,\sigma }E_{n,m,\sigma }^{l}P_{n,m,\sigma }\left(T_{l},B_{h}\right)+U_{D}\left(T_{l},B_{h}\right)-\sum_{n,m,\sigma }E_{n,m,\sigma }^{h}P_{n,m,\sigma }\left(T_{h},B_{l}\right),$$
where, due to the thermal equilibrium of the two points (Points D and B in Figure 3), we can define the internal energy from equilibrium partition function. If we subtract the classical work given by Equation (Equation 23) from the quantum work written in the form of Equation (Equation 27), we obtain the following equation
(28)$$W-W=\left(\sum_{n,m,\sigma }E_{n,m,\sigma }^{l}P_{n,m,\sigma }\left(T_{l},B_{h}\right)-U_{A}\left(T_{A},B_{l}\right)\right)+\left(\sum_{n,m,\sigma }E_{n,m,\sigma }^{h}P_{n,m,\sigma }\left(T_{h},B_{l}\right)-U_{C}\left(T_{C},B_{h}\right)\right)$$

The first summation of Equation (Equation 28) is the average of the energy at low magnetic field with thermal probabilities that satisfies the adiabatic condition
(29)$$S=-k_{B}\sum_{n,m,\sigma }P_{n,m,\sigma }\left(T_{l},B_{h}\right)\ln\left(P_{n,m,\sigma }\left(T_{l},B_{h}\right)\right),$$
i.e., the entropy at Point A. On the other hand, $U_{A}\left(T_{A},B_{l}\right)$ is the average value of the energy at low external field and in thermal equilibrium, with the same value of entropy presented in Equation (Equation 29). Therefore, $U_{A}\left(T_{A},B_{l}\right)$ is the absolute minimum according to thermodynamic [53]. The same argument can be made at Point C, thus the difference of classical work minus quantum work always satisfies the following condition
(30)W−W≥0

This result applies to any system in which the occupation probabilities of the energy levels at any magnetic field are replaced with any form, provided that they satisfy the adiabatic condition. This is because the value at equilibrium of any internal parameter (without constrains) of the system, makes the internal energy to be a minimum for a given value of the total Entropy [53].

## 5. Conclusions

In this work, we explored the classical and quantum approach for a magnetic Otto cycle for an ensemble of non-interacting electrons with intrinsic spin where each one is trapped inside a semiconductor quantum dot modeled by a parabolic potential. We analyzed all relevant thermodynamics quantities, and found that the entropy changes it behavior in terms of the external magnetic field at the point where the energy spectrum tends towards degeneracy; this behavior was present at all temperatures considered. This behavior determined the range of parameters such as temperature and external magnetic field that would lead to the operation of the Otto cycle extracting positive total work. In the classical approach, we found oscillations in the total work extracted that are not present in the quantum approach. This happened near the zone of slope change in the behavior of the entropy in terms of the magnetic field. Interestingly, we found that, in the classical approach, the engine yielded a much higher performance in terms of total work extracted and efficiency than in the quantum approach. This is because, in the classical approach, the working substance can be in thermal equilibrium at each point of the cycle, whereas, in the quantum approach, the working substance can only thermalize in the isochoric strokes. Because of the principle of minimum energy, the system is allowed to extract more energy when the adiabatic strokes can lead to states that are in thermal equilibrium, which is only possible in the classical case.

These results are reasonable, since, in our quantum approach, the working substance remains in a diagonal state and does not use quantum resources such as quantum coherence, which in some cases can lead to enhanced performance.

## Figures and Tables

**Figure 1 entropy-21-00512-f001:**
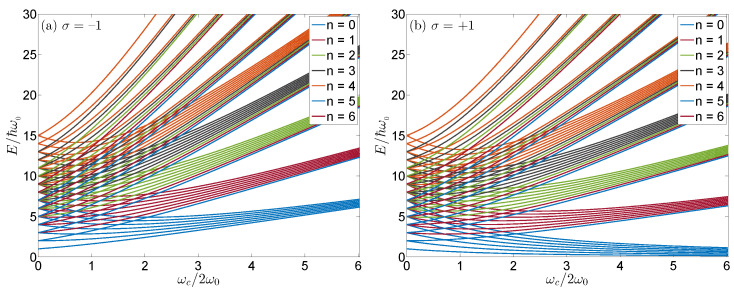
(**a**) Fock–Darwin energy spectrum with σ=−1 for the first six radial number n=0,1,…,6 and for each of them the azimuthal quantum number taking the values between m=−6,−5,…,5,6. (**b**) Fock–Darwin energy spectrum with σ=+1 for the first six radial number n=0,1,…,6 and for each of them the azimuthal quantum number taking the values between m=−6,−5,…,5,6. We clearly observe the confinement of the energy levels at high magnetic fields $\left(\omega_{c}/2\omega_{0}>>1\right)$.

**Figure 2 entropy-21-00512-f002:**
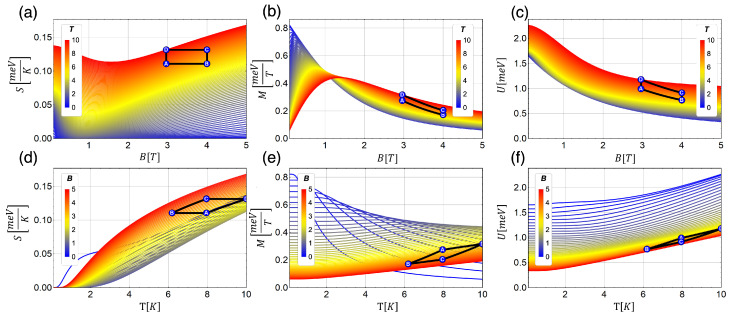
Classical thermodynamic quantities entropy (S), internal energy (U) and magnetization (M) as a function of: external magnetic field (*B*) (**a**–**c**); and temperature (*T*) (**d**–**f**). In (**a**–**c**), the colors blue to red represent temperatures from 0.1 K to 10 K, respectively. For (**d**–**f**), the colors blue to red represent lower to higher external magnetic field, from 0.1 T to 5 T. The value of the parabolic trap is approximately to 1.7 meV. Additionally, we show how the Otto cycle appears in terms of the thermodynamic quantities considered.

**Figure 3 entropy-21-00512-f003:**
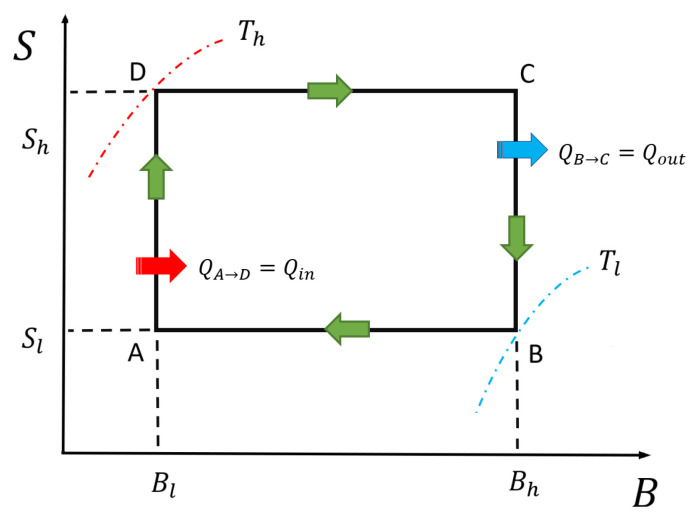
The magnetic Otto engine represented as an entropy (S) versus a magnetic field (B) diagram. The way to perform the cycle is in the form B→A→D→C→B.

**Figure 4 entropy-21-00512-f004:**
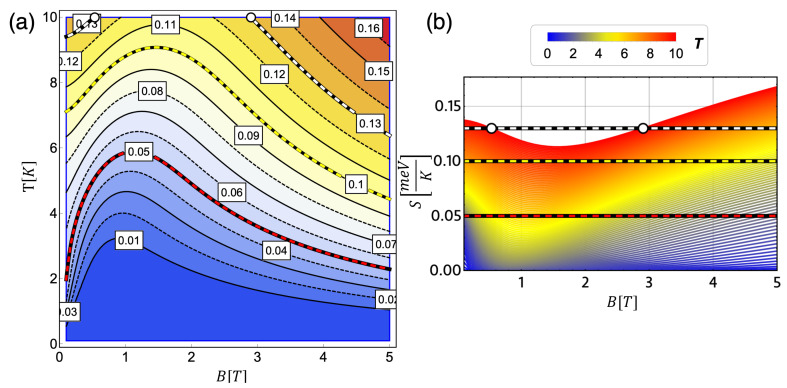
Solution of classical isentropic path. (**a**) The entropy as a function of magnetic field (horizontal axis) and temperature (vertical axis). The level curves (constant entropy values) highlight three different cases for *S*: first, red-black curve corresponding to S=0.05; secondly, yellow-black curve, corresponding to S=0.10 and finally, white-black curve for the case of S=0.13. (**b**) The three constant values for the entropy (S=0.05,S=0.10,S=0.13) in a graphic of entropy as a function of *B* for temperatures from 1 K (blue) up to 10 K (red). Due to the form of the entropy obtained for this system, the solution for S=0.13 needs to work with temperatures higher than 10 K for an external magnetic field lower than 3 T (white dots in (**a**,**b**)). The value of the parabolic trap corresponds to 1.7 meV.

**Figure 5 entropy-21-00512-f005:**
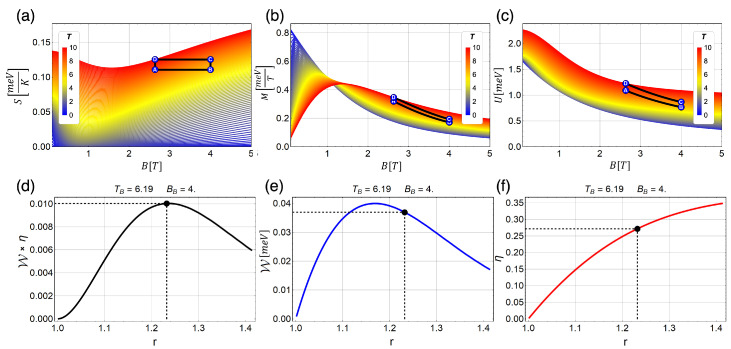
Proposed magnetic Otto cycle showing three different thermodynamic quantities, Entropy (*S*), Magnetization (*M*) and Internal Energy (*U*) ((**a**–**c**), respectively) as a function of the external magnetic field and different temperatures from 0.1*K* (blue) to 10*K* (red). (**d**) The total work extracted multiplied by efficiency (Wη); (**e**) the total work extracted (W); and (**f**) the efficiency (η) for the classical cycle. The black points in (**d**–**f**) represent exactly the cycle B → A → D → C → B, presented in (**a**–**c**). The value of the parabolic trap corresponds to 1.7 meV. The fixed temperatures are $T_{B}=6.19$ K and $T_{D}=10$ K.

**Figure 6 entropy-21-00512-f006:**
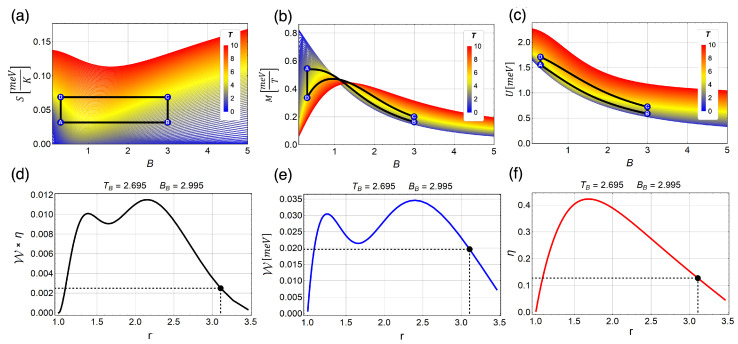
Proposed magnetic Otto cycle in three different thermodynamics quantities, Entropy, Magnetization and internal energy ((**a**–**c**), respectively) as a function of the external magnetic field and different temperatures from 0.1 *K* (blue) to 10 *K* (red). Total work extracted multiplied by efficiency (Wη) (**d**) total work extracted (W) (**e**) and efficiency (η) (**d**) for the cycle. The black point in (**d**–**f**) represents the value of 0.02 meV of total work extracted and corresponds exactly to the cycle B → A → D → C → B, shown in (**a**–**c**). The value of the parabolic trap correspond to 1.7 meV. The fixed temperatures are $T_{B}=2.69$ K and $T_{D}=5.40$ K.

**Figure 7 entropy-21-00512-f007:**
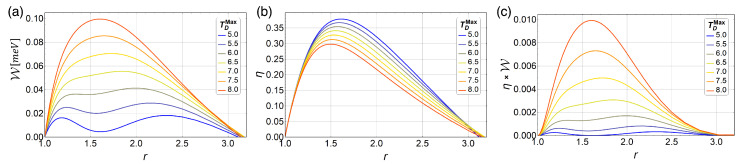
Work, efficiency and work multiply by efficiency (**a**–**c**) for different values of $T_{D}$ for $T_{B}=2.69$ fixed. The value of the parabolic trap corresponds to 1.7 meV.

**Figure 8 entropy-21-00512-f008:**
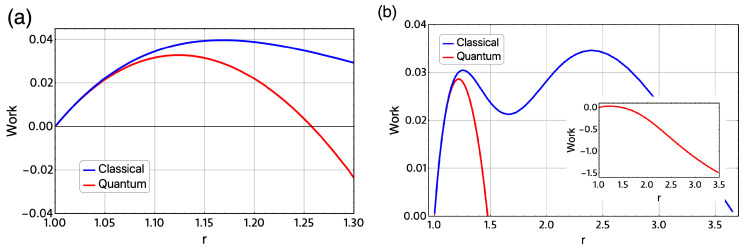
(**a**) Total work extracted for classical (blue line) and quantum version of Otto cycle (red line). The parameters for this case displayed are: $T_{D}=10$ K, $T_{B}=6.19$ K and $B_{B}=4$ T as starting value of the external magnetic field. The value of $B_{D}$ moves from 4 T to 1.99 T and this variation is reflected in the movement of *r* in the form of $r=\sqrt{\frac{4}{B_{D}}}$, same parameter as the results shown in Figure 5. (**b**) Total work extracted (W) presented in Figure 6e versus the values obtaining in the quantum version of the Otto cycle. The parameters for this figure are $T_{D}=5.40\;K$, $T_{B}=2.69\;K$ and $B_{B}=2.995$ T and $B_{D}$ moves from 2.995 to 0.250 T. The parabolic trap is fixed to the value of 1.7 meV and the effective mass value of $m^{*}=0.067m_{e}$.

**Figure 9 entropy-21-00512-f009:**
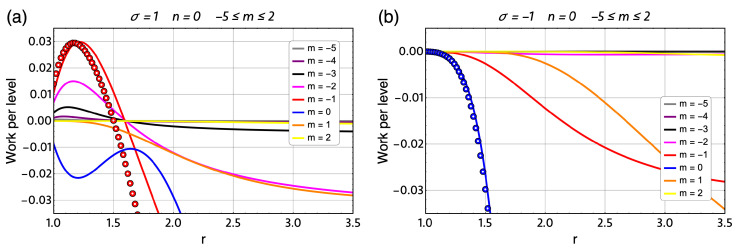
Total quantum work extracted (W) per energy level for the case of σ=1 (**a**) and for the case of σ=−1 (**b**). The lines marked with circles correspond to the sum of all contributions of the energy level for each spin. The parameters used for this figure are the same as the one used in Figure 8b.

**Figure 10 entropy-21-00512-f010:**
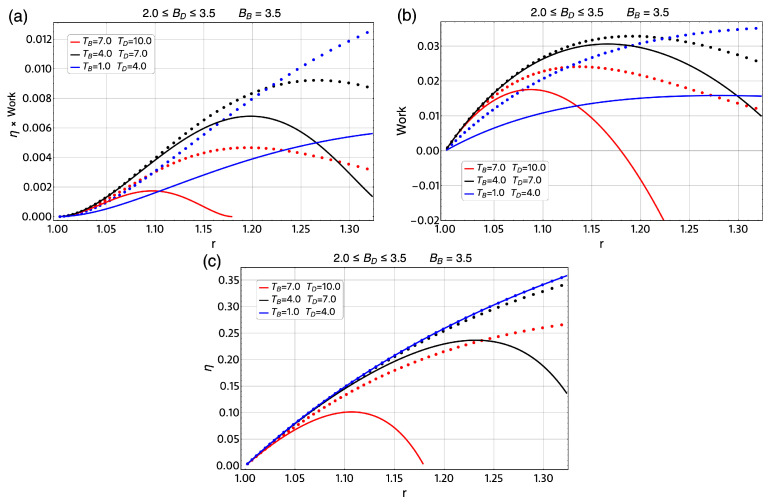
η×W (**a**); and total work extracted (**b**,**c**) efficiency for the case of $\Delta T=T_{h}-T_{l}=3\;K$ for different regions of temperature parameter for classical approach (solid line) and quantum version of the magnetic Otto cycle (dotted line). For all graphics, we use the initial external magnetic field in the value of $B_{B}=3.5$ T and the minimum value of the field, $B_{D}$ moves between 3.5 T and 2.0 T. Therefore, the *r* parameter moves between 1≤r≤1.32. The parabolic trap is fixed to the value of 1.7 meV and the effective mass value of $m^{*}=0.067m_{e}$.

**Figure 11 entropy-21-00512-f011:**
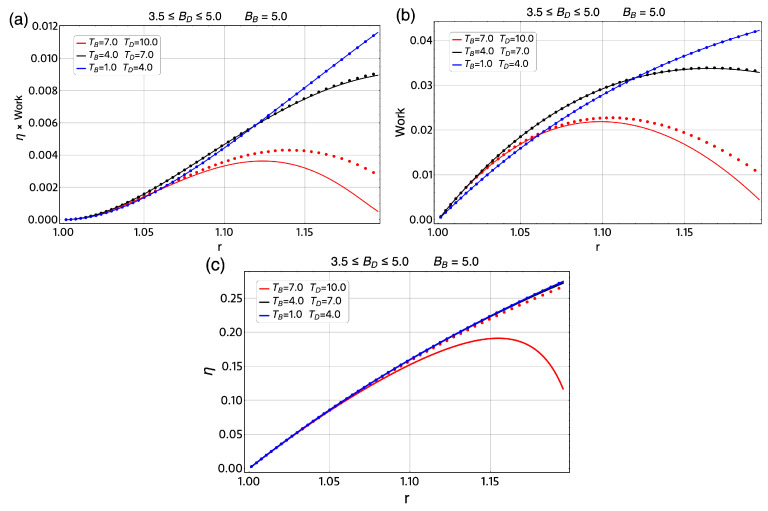
η×W (**a**); and total work extracted (**b**,**c**) efficiency for the case of $\Delta T=T_{h}-T_{l}=3\;K$ for different regions of temperature parameter. For all cases, we use the initial external magnetic field at the value of $B_{B}=5.0$ T and the minimum value of the field, $B_{D}$ moves between 5.0 T and 3.5 T. Therefore, the *r* parameter moves between 1≤r≤1.19. The parabolic trap is fixed to the value of 1.7 meV and the effective mass value of $m^{*}=0.067m_{e}$.

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
