# Peer review of "Magnetic Otto Engine for an Electron in a Quantum Dot: Classical and Quantum Approach"

_entropy, 2019, doi:10.3390/e21050512_

Round 1
Reviewer 1 Report
The authors investigated the performance of a quantum magnetic Otto cycle working with the Fock Darwin model. Staring frorm the expression and thus the partition function of the working substance, they present numberical calculations for the thermodyannic quantities such as the entropy, efficiency and work, etc., and their results show the effects of the system paramters on the the engine performance. The authors claimed that the quantum Otto engines have larger work and efficiency than their classical counterparts. The results are meaning. Hower, the authors just focused on the numerical calculations, without giving physical mecahinsms to convince the readers. Before clarifying this point, I cannot recommend the present paper for publication in Entropy.
Author Response
Dear Referee, the answers to your comments can be found in the attached letter.
Best Regards,
Francisco J. Peña on behalf all authors.

Reviewer 2 Report
The paper by Pena et al. discusses an Otto cycle for a system with a working fluid modeled by the Fock-Darwin model. Thair main finding is that when computing the work output for a quanutm or a classical cycle they can get qualitatively different behaviors in a certain parameters region. Also they show that the classical cycle outperforms the quantum one. This is fairly well explained and quite interesting. So I think the paper could be accepted, however I would ask the authors to consider the comments below.
1) The authors cite [30] as a quantum Otto cycle, but this is not true. Although it is a cycle made with just one atom, it is still a classical experiment, and there is no controversy or doubt about it. The heating is classical noise, and the occupation of the phonon modes is of thousands. There are instead quantum engines cycles, and for them you could cite
- J. Klatzow, J. N. Becker, P. M. Ledingham, C. Weinzetl, K. T. Kaczmarek, D. J. Saunders, J. Nunn, I. A. Walmsley, R. Uzdin, and E. Poem, arXiv:1710.08716 (2017).
- J. P. Peterson, T. B. Batalhão, M. Herrera, A. M. Souza, R. S. Sarthour, I. S. Oliveira, and R. M. Serra, arXiv:1803.06021 (2018).
- D. von Lindenfels, O. Gräb, C. T. Schmiegelow, V. Kaushal, J. Schulz, F. Schmidt-Kaler, and U. G. Poschinger, arXiv:1808.02390 (2018).
- N. Van Horne, D. Yum, T. Dutta, P. Hänggi, J. Gong, D. Poletti, and M. Mukherjee, arXiv:1812.01303 (2018).
2) I do not understand why the authors write that the expression for the efficiency of an Otto cycle is "in first approximation ... depends on the quotient of the two temperatures". For the classical Otto cycle with an ideal gas, this is exactly true. Are you trying to write that this is the classical limit? I don't know, I find the "first approximation" too vague.
3) I find the introduction not so linear, and I would ask the authors to make it more concise. It seems to me that they are trying to write too much.
4) It would be good to have a color bar for Figs. 2, 4, 5, 6, 7.
5) It would also be good to have units for all quantities.
6) Line 152, do the authors mean point C instead of point D?
7) It could help the reader to use a different notation for the classical net work and the quantum one. Like W_c and W_q.
8) Equation (14) is very dangerous. What the authors write for work, although found in the literature, may not be correct, and in general it contradicts the two-time measurement protocol of Hanggi, Lutz, Talkner. In fact in case the system is at t=0, in a superposition of different energy eigenomodes, then the first energy measurement will dastrically change the system and the value of work will be different from that of Equation (14). If I am not wrong, fortunately for the authors, they always deal with density matrix of the system which is diagonal in the energy eigenbasis.
9) In Fig.8(a) what is crossing 0 at r~1.36? The classical one which is not shown? Or the quantum one which actually crosses at 1.26? I am confused.
10) In various parts of the papers the authors refer to an infinite degeneracy. But in their work they only use finite m. So I find this presentation a bit confusing.
Author Response

(The authors gave the same response as above.)

Reviewer 3 Report
This paper describes in some detail the use of a semiconductor quantum dot to perform a thermodynamic Otto cycle, obtaining a thermal engine operating in the quantum regime.
The behaviour of this thermal machine, externally controlled via a magnetic field, is analyzed in various regimes, using the Fock-Darwin model for the dot and obtaining interesting results and a variety of different behaviours depending on the working point.
In particular, very interestingly, the authors compare the classical description with the full quantum one, showing that the latter gives rise to a lower efficiency. In my opinion, this is one of the more important merits of the paper, which, perhaps, would deserve some more emphasis in the introductory section, as such a quantum vs classical description is becoming the focus of a specific research line (see e.g. Brandner et al, PRL 119, 170602 (2017) or Pekola et al., arXiv:1812.10933 and references therein).
In this respect, besides introducing the question in a more suitable way, I'd suggest the authors to try and make an effort to provide a physical interpretation of this result, which is possibly due to the presence of coherence in a finite time quantum cycle as argued in the previously mentioned papers or to the production of quantum inner friction (as discussed in some of the papers by R. Kossloff cited in this manuscript, or in Alecce et al., New J. Phys. 17, 075007 (2015) ).
Summarizing, I suggest the acceptance of the manuscript after the authors have made these changes (there is no need for a further review).
Some further, minor points are the following
- in the caption of Fig 1, the letter s is used for the spin , rather than the \sigma adopted everywhere else;
- in Fig. 2, a color code would be useful
- in the caption of Fig. 2 it seems that the descriptions of panels (a)-(c) and (d)-(f) have been interchanged
- a general spell check may be in order.
Author Response

(The authors gave the same response as above.)

Round 2
Reviewer 1 Report
The authors have revised the paper according to my suggestion. I am satisfied with the current manuscript. However, a minor point that the thermal equlibrium must be under the condition of the quasi-static limit should be clarified before its publication in Entropy.